# The Antimicrobial and Antibiofilm Abilities of Fish Oil Derived Polyunsaturated Fatty Acids and Manuka Honey

**DOI:** 10.3390/microorganisms12040778

**Published:** 2024-04-11

**Authors:** Jenna Clare, Martin R. Lindley, Elizabeth Ratcliffe

**Affiliations:** 1Department of Chemical Engineering, Loughborough University, Loughborough LE11 3TU, UK; 2School of Health Sciences, Faculty of Medicine and Health, University of New South Wales, Sydney 2052, Australia; m.lindley@unsw.edu.au

**Keywords:** docosahexaenoic acid, eicosapentaenoic acid, biofilm, ESKAPE, manuka honey

## Abstract

Both honey and fish oil have been historically used in medicine and identified as having antimicrobial properties. Although analyses of the substances have identified different components within them, it is not fully understood how these components interact and contribute to the observed effect. With the increase in multi-drug resistant strains of bacteria found in infections, new treatment options are needed. This study aimed to assess the antimicrobial abilities of fish oil components, including docosahexaenoic acid (DHA), eicosapentaenoic acid (EPA), and derived resolvins (RvE1, RvD2, and RvD3), as well as two varieties of manuka honey, against a panel of medically relevant microorganisms and antimicrobial resistant organisms, such as Methicillin Resistant *Staphylococcus aureus* (MRSA) and carbapenem-resistant *Escherichia coli*. Minimum inhibitory concentrations (MIC) and minimum bactericidal concentrations (MBC) were identified; further minimum biofilm eradication concentrations (MBEC) were investigated for responsive organisms, including *S. aureus*, *E. coli*, *Staphylococcus epidermidis*, *Klebsiella pneumoniae*, and *Pseudomonas aeruginosa.* Concurrent with the existing literature, manuka honey was found to be a broad-spectrum antimicrobial with varied potency according to methylglyoxal content. DHA and EPA were both effective against Gram-positive and negative bacteria, but some drug-resistant strains or pathogens were not protected by a capsule. Only *E. coli* was inhibited by the resolvins.

## 1. Introduction

The use of natural substances throughout medicine has a long history [1], but such treatments began to fall out of favour as newer advances have been made. Although its antimicrobial properties were first properly observed in 1892, honey has been used medicinally for centuries, with references dating back as far as the Old Testament of the Bible and the Quran [2]. Cod liver oil has been taken by people for many years, and it has remained a popular supplement due to its reported positive effects [3]. In the twentieth century, paediatricians began to recommend that children receive regular doses of it for their general health as a preventative action [1].

Fish oil is comprised of a range of omega fatty acids that humans are not able to synthesise and must be acquired through dietary sources [4]. These are classed as essential fatty acids due to their presence in the cell membrane and abundance in the brain, retina, and neural tissues [5]. They are metabolised by the body; both eicosapentaenoic acid (EPA) and docosahexaenoic acid (DHA) are readily available omega-3 fatty acids. From these, the human body synthesises resolvins (Rvs) (Figure 1). E-series resolvins (denoted as RvE) are derived from EPA, and D-series resolvins (denoted as RvD) are derived from DHA [6]. DHA is converted to 17S-hydroperoxy-DHA by 15-lipoxygenase, and this intermediate is converted into one of six discovered resolvins. EPA is converted by COX-2 into 18R-hydroperoxy-EPA. This intermediate is also converted into one of four resolvins via an epoxidation reaction [7]. Resolvins are produced throughout the resolution stage of inflammation, which is where they gain their name from [8]. 

EPA and DHA have demonstrated previous antimicrobial and antibiofilm capabilities in eradicating *Porphyromonas gingivalis*, *Fusobacterium nucleatum* [9], and *Streptococcus mutans* [10]. Other organisms with known sensitivities to EPA include *Bacillus cereus* [10], *Klebsiella pneumoniae* [11], and *S. aureus* [12]. Previous work also suggests that the bacteria gained no resistance to the treatment after 13 passages in sub-inhibitory concentrations of EPA [12], indicating that these fatty acids could be of particular use given the rise in resistance to traditional antibiotics. Recurrent worldwide trends in antibiotic resistance have led to the establishment of the ESKAPE pathogen list. ESKAPE is an acronym that stands for the six organisms that have been identified as having high levels of associated mortality, have multi-drug resistance (MDR), and require new therapeutics. These pathogens are *Enterococcus faecium*, *S. aureus*, *K. pneumoniae*, *Acinetobacter baumannii*, *P. aeruginosa*, and *Enterobacter* spp. [13]. There is evidence to suggest that EPA and DHA are effective against ESKAPE pathogens, showing they are worth further investigation.

Honey has multiple elements that contribute to antimicrobial effects, but what sets manuka honey apart is its methylglyoxal (MGO) content. MGO has been identified as a major contributor to antimicrobial abilities [14]. Manuka honey is available widely and often sold according to its MGO content, which varies from 50 mg/kg to 800+ mg/kg. It has been previously suggested that the quantity of MGO in a honey sample is dependent on the soil fertility in the range of the foraging bees and that this may account for the variety of MGO content [15]. In previous studies, a higher MGO content has led to a greater antimicrobial effect [16], and the antimicrobial effect of manuka honey has been replicated using a dedicated MGO solution [17]. Although the exact mechanism of action of MGO is not fully understood, numerous previous studies have documented successful bacterial death and inhibition of species such as *E. coli*, *P. aeruginosa*, and MRSA [18,19]. In some instances, honey has already been used in a clinical setting and has been successful in dressing diabetic foot ulcers when combined with antibiotics. Although the microbial response to the honey dressing was mixed, *Enterococcus* spp. and *Pseudomonas* spp. load was reduced compared to the control group. *Enterococcus* and *Bacteroides* persisted, but the patients also reported less pain, less oedema, and reduced odour [20]. An additional case study found that a persistent 3-year-old chronic wound healed after combining manuka honey with amoxiclav, and after 7 days of treatment, a swab found no bacteria, and recurrent infections ceased [21].

The issue of increasing antibiotic resistance has been well documented [22], and it is expected that by 2050, 10 million deaths will be attributed to antibiotic resistance [23]. Several species of bacteria have been identified as particularly problematic as they already have high levels of associated mortality and are resistant to multiple drugs; these are known as ESKAPE pathogens and include *Enterococcus faecium*, *S. aureus*, *K. pneumoniae*, *Acinetobacter baumannii*, *P. aeruginosa*, and *Enterobacter* spp. [13]. Infections can often lead to the formation of biofilms, characterised by slimy layers or aggregates which can attach to surfaces—up to 80% of bacterial infections in the body have biofilms present [24]. After adhesion, biofilms begin to mature and form aggregates and matrices, which can lend them resistant properties not seen by motile bacteria, and for this reason, the prevention of biofilm development has also become of greater interest in treatment options [25], with novel antibiotics raised as a potential solution to this issue [26]. The nature of biofilms makes them more difficult to treat than planktonic bacteria, as evidence shows they may require a stronger concentration of antibiotics for treatment [27]. ESKPAE pathogens are all capable of forming biofilms, and the combination of potential multi-drug resistance makes treating them more difficult [13]. As biofilms can also be polymicrobial, it is important to identify treatments that are broad spectrum, and early eradication before maturation may aid in the prevention of more difficult polymicrobial biofilms [25].

This study aimed to assess the antimicrobial ability of different treatments systematically. This included inhibition and bactericidal measurements, as well as the ability to reduce viable cells in biofilms. Fish oil components to be tested were DHA and EPA, and they subsequently produced resolvins RvE1, RvD2, and RvD3. Two varieties of manuka honey were also used, as well as a synthetic MGO solution to compare to the honey. While there are existing screens of manuka honey, the data for EPA and DHA are currently much more limited when considering the breadth of bacteria tested. Honey was included so as to be able to compare to existing work when developing the biofilm work for the fish oils. Knowledge of novel antimicrobials will aid in the treatment of bacterial infections and may also help to prolong the useful period of antibiotic treatments that, due to the rise of antibiotic resistance, are beginning to become less effective.

## 2. Materials and Methods

### 2.1. Materials

All broth and agar powders were obtained from Merck (Gillingham, UK). RvD2, RvD3, RvE1, EPA, and DHA were purchased from Cayman Chemical (all ≥98% purity) (Cambridge, UK). ATCC 25922 *E. coli*, ATCC14990 *S. epidermidis*, and MRSA 252 were all obtained from the onsite existing stock at Loughborough University. ATCC6538 *S. aureus*, ATCC27853 *P. aeruginosa*, and ATCC13883 *K. pneumoniae* were sourced from ATCC (Middlesex, UK). Carbapenem-resistant *E. coli* (CRE) was obtained from the Queens Medical Centre Pathogen Bank (Nottingham, UK).

‘Pure Gold Manuka Honey 850+ MGO’ was purchased from Holland & Barrett (Nuneaton, UK); ‘Activon Tube’ is widely available from chemists but, in this instance, was purchased through amazon.co.uk. MGO solution was purchased from Merck (Gillingham, UK).

The BacTiter-Glo Microbial Cell Viability kit was purchased from Promega (Southampton, UK).

### 2.2. Determination of the Methylglyoxal (MGO) Concentration of Manuka Honey Samples

Although the MGO content of the pure gold honey is understood to be 850+, the MGO content of the medical-grade honey was not provided by the manufacturer. However, upon further investigation using the methylglyoxal assay kit (Abcam, Cambridge, UK), this was determined to be 130 MGO. The pure gold honey was also investigated and determined to be 890 MGO.

### 2.3. Minimum Inhibitory Concentration (MIC) and Minimum Bactericidal Concentration (MBC)

Methods were adapted from the standard protocols defined by the European Committee on Antimicrobial Susceptibility Testing (EUCAST) (EUCAST, 2003) [28]. Bacteria were incubated at 37 °C in a shaking incubator at 225 rpm. Prior to experimental work, 24 h standardised growth curves were produced to ensure that the bacterial concentration applied to this study was consistent. Mueller Hinton broth and agar were used throughout.

In brief, bacteria were cultured to a concentration of 1.5 × 10^8^ colony forming units (CFU)/mL, and 100 µL was added to a 96-well plate (1.5 × 10^7^ cells per well). DHA and EPA were diluted in phosphate-buffered saline (PBS) at concentrations of 12 mM and 2-fold serially diluted. Resolvins were all prepared to 200 nM and 2-fold serially diluted. Each honey was diluted in PBS to 75% (*w*/*v*) and 50% (*w*/*v*), the second of which was also 2-fold serially diluted. Synthetic MGO was diluted in PBS to make solutions equivalent to pure gold and medical-grade honey. These were then used to prepare diluted stocks in the same concentrations as the complete honey. A total of 100 µL of treatments were also added to the wells with the bacteria. The absorbance of the plate was measured at 600 nm using the FLUOStar Omega microplate reader (BMG Labtech, Ortenberg, Germany). The plate was then incubated at 37 °C for 18 h, after which the absorbance was measured again.

To determine the point of the MIC, a one-way ANOVA was performed with Tukey post hoc analysis. The MIC was the point at which there was no significant difference between the treated sample and the measured absorbance of the bacterial sample at the point of inoculation. A treatment was described as having a partial inhibitory effect if the absorbance was significantly different from both the inoculation and the untreated bacterial control.

To assess the MBC, 15 µL of each sample was transferred from each well of the well-plate after taking the final absorbance measurement and spotted onto agar plates, which were prepared in duplicate. Agar plates were incubated for 24 h at 37 °C. The MBC was determined visually and was the point at which the transferred sample failed to grow on agar.

### 2.4. Minimum Biofilm Eradication Concentration (MBEC)

Biofilms were established by adding 100 µL (equal to 1.5 × 10^7^ cells per well) of cultured bacteria to a 96-well plate and incubating statically at 37 °C. After 24 h, the media was aspirated from the plate, which was then gently rinsed twice with PBS to remove any unattached cells. A total of 100 μL of fresh media and 100 μL of treatment solution was added to the plate, which was returned to the incubator for a further 24 h. The plate was washed with PBS again and then placed in the Transsonic T480 (Camlab, Cambridge, UK) water bath for 10 min at 35 Hz to detach biofilm cells.

The ATP reagents from the BacTiter-Glo kit were prepared according to the manufacturer’s instructions. The solution causes cell lysis, releasing the ATP from the cells. The kit contains luciferase, which emits luminescence in the presence of ATP and can be measured with the plate reader. A total of 100 μL of ATP reaction mix was added to 100 μL of the cell solution detached from the well plate and transferred to a dark-walled plate. After 5 min of incubation at room temperature in the dark, the luminescence of the plate was measured using the plate reader. Wells that were below the detection limits were plated onto agar to confirm the presence of any viable cells.

Only organisms that were deemed to have a bactericidal response to treatment during MIC and MBC investigations proceeded to the MBEC investigation, with the additions of MRSA and *E. coli* used as representative Gram-positive and negative organisms, respectively. Additionally, *K. pneumoniae* was included due to the suggestion in the existing literature that DHA treatment of these biofilms leads to an increase in growth [29]. Pure gold honey proceeded to the MBEC investigation as a treatment option, whereas the medical-grade variety was not continued as it was shown to not produce a bactericidal effect.

## 3. Results

A full panel of results for the MIC, MBC, and MBEC is shown in Table 1.

Figure 2 is included as an indication of the MIC-measured absorbances of MRSA and demonstrates how the treatments affected bacterial growth as observed by the measured absorbance. Inhibited treatments show the absorbance was not significantly different from the starting inoculation value—the density of the solution remained unchanged, and the CFU remained low. Treatments where the absorbance was between the inoculation and untreated control were classed as partially inhibited—where the treatment failed to completely inhibit growth yet prevented the growth from reaching where it would without any treatment.

DHA and EPA were effective treatments against *E. coli*, *S. aureus*, *S. epidermidis,* and MRSA. EPA treatment led to a partial inhibition of *P. aeruginosa* and *K. pneumoniae*; DHA also partially inhibited the growth of *P. aeruginosa* but failed to have an impact on *K. pneumoniae.* Table 1 also shows that EPA was a more effective treatment as it largely required less concentrated treatments to achieve inhibition and cell death; EPA achieved bactericidal responses at 1.50–3 µM, compared to the 3–12 µM recorded for DHA.

The resolvins were much less effective and, in most cases, achieved only partial inhibition of the bacteria; *E. coli* was successfully inhibited at concentrations of 125 nM. This supports the findings of Norris et al. (2018) [30], who also found that the D-series resolvins were capable of *E. coli* inhibition. Due to the lack of complete inhibition for the other bacterial strains, the resolvins, and other less effective treatments were not continued in the MBC and MBEC sections of this study.

Both brands of manuka honey used were effective inhibitory agents at concentrations from 12.5 to 75% (*w*/*v*). However, only the pure gold honey had a complete bactericidal action against all of the pathogens; the medical-grade honey only proved to be bactericidal against *S. epidermidis*. It would, therefore, be expected that the pure gold honey had a higher antimicrobial effect, which is reflected in the results. Table 1 shows that aside from the medical-grade honey MIC for *E. coli*, all recorded MICs and MBCs were equivalent to the complete honey.

Pure gold honey was effective as an antibiofilm treatment against all the organisms tested at 50–75% (*w*/*v*). DHA and EPA were both effective against MRSA and *E. coli*, although EPA was required at smaller concentrations of 6 mM compared to the 24 mM of DHA required to produce the same effect. *K. penumoniae* was not eradicated by either of the fish oil treatments, though this was not unexpected based on its lack of MBC. It was included in this study as the previous literature [29] suggested that DHA can cause an increase in growth in *K. pneumoniae* biofilms. This was not reflected here—the biofilm was not eradicated, but there were significantly fewer viable cells after treatment with all doses of EPA or DHA than with an untreated biofilm. The ATP assay indicated a significant 2-fold reduction in viable cells (*p* < 0.05). This is important to establish when considering treatments that may be suitable for further clinically relevant research, especially as *K. pneumonia* is known to be prevalent in a range of wounds, including burns [31], diabetic foot ulcers [32], and surgical site infections [33].

## 4. Discussion

DHA has previously been shown to inhibit *S. aureus* and *E. coli* [34], although Shin et al. reported that *K. pneumoniae* and *P. aeruginosa* were also inhibited by both DHA and EPA, which contrasts what has been reported here, where only a partial inhibition was achieved for *P. aeruginosa*. Differences are potentially due to experimental design and source of fish oil.

Interestingly, the carbapenem-resistant *E. coli* (CRE) was not fully inhibited by any of the fish oil components, whereas the lab strain of *E. coli* was inhibited at the lowest doses of fish oils tested. This may suggest that the resistance mechanisms gained by the CRE help it to resist the fish oils. The results mean that fish oil treatments may not always be appropriate as a clinical antimicrobial, depending on the pathogens present, and that a bacterial screen may be required to judge efficacy before application. This suggests that further work is required in investigating a larger variety of drug-resistant strains, as beyond MRSA, there are very few included in many studies, yet systematic data will be critical in working against the rise of multi-drug-resistant bacteria.

The previous literature also suggests that EPA and DHA can treat biofilms, although these papers focus on other bacteria, including *F. nucleatum* and *P. gingivalis* [9]. The concentrations required were significantly lower (200 µM), but this study focused primarily on the prevention of biofilm formation rather than biofilms that were forming or already established. The bacteria used may also be less virulent than ESKAPE pathogens and not require such high doses to treat. This work supports that some bacteria are susceptible to fish oil treatments but also demonstrates that not all organisms are killed, as shown with *K. pneumoniae*.

In the case of EPA, 6 mM was enough to eradicate *E. coli* and MRSA biofilms; this was only one 2-fold dilution step above the MBC for *E. coli* and two for MRSA (MBC was 1.5 and 3 mM, respectively). Following the existing trend, DHA required a higher concentration, and 24 mM was sufficient in both cases. This again followed the same pattern as with EPA, where *E. coli* required a dose of one 2-fold dilution higher, and MRSA needed two. Biofilms often require significantly higher doses of antibiotics to treat; these results may suggest that the treatments encounter little issues when crossing the slimy layer of the biofilm. This has positive implications as high concentrations of treatments can cost substantially more and have the potential to reach concentrations that may show patient toxicity.

Figure 3 shows an unexpected trend that occurred when treating *P. aeruginosa* with honey. The middling concentrations caused an increase in growth past the untreated control, which was not observed at the lowest concentrations tested. This was only observed in the species of bacteria in this study. There is existing evidence that diluted manuka honey can cause an increase in bacterial growth to a point higher than if untreated [35]. A concentration of 2.5% (*w*/*v*) manuka honey solution caused a significant increase in the total viable cell count of MRSA over 24 h. Although this was not seen with MRSA in this study, it has occurred with *P. aeruginosa*. The authors also do not suggest a reason for this. It was considered whether the sugar content of the honey outmatched the low MGO of the solution and encouraged growth; however, other authors have prepared a sugar control comprised of fructose, glucose, maltose, and glucose alongside dilutions of honey [36]. The sugar solutions still led to a reduction in the growth of *P. aeruginosa* compared to the control; therefore, it is unlikely that the high sugar content providing an extra nutritional source was the causative agent.

As the accelerated growth was seen only in *P. aeruginosa,* it is possible that there is something unique about the bacteria that causes the effect. *E. coli* does show a similar but less extreme rate of growth for the middle concentrations, though these are not significantly different from the untreated control. As there are other Gram-negative and other capsuled bacteria that have not shown a 2-fold increase in growth compared to the control, the capsule or Gram status is less likely to be the cause.

Direct comparison to other studies using manuka honey is difficult due to the large variety of manuka honey that has been used by different authors. MGO is not always reported, but there is a general trend of manuka honey studies with positive antimicrobial responses [19,37], which is further supported by this work. Although both kinds of honey have shown positive effects, and the MGO solution results closely mirror the honey results, the contribution of MGO to the antimicrobial effect cannot be attributed solely to MGO as other antimicrobial compounds exist within most honey varieties, and these have not been investigated as part of this study.

These types of studies suggest that EPA and DHA make effective treatments in preventing biofilm growth, which would be particularly advantageous should further research identify that bacteria do not develop resistance mechanisms against them. Though sometimes unavoidable in a prophylaxis case, using antibiotics as preventative treatments is generally discouraged due to the promotion of AMR it can cause, but preventing biofilm formation in the first place would be better than attempting to eradicate once already formed, as it could reduce the likelihood of further wound treatment costs and improve patient experience.

Alternative treatments to traditional antibiotics are of increasing importance, and this systematic analysis shows that these substances may be appropriate and should be considered for further research. However, it is important to note that they may not be appropriate for all patients—those with bee sting allergies may not be suitable recipients of a honey treatment, and it has been suggested that diabetic patients may also not be suitable due to the application of a high sugar product, though this has been disputed [38]. Although the fish oil components can be synthesised from organic components [39], these remain very expensive compared to those extracted from fish oil, which could be rejected by some patients on an ethical basis in the case of vegetarians and vegans. Future work should focus on elucidating the mechanism of actions for all treatments, which are yet to be fully understood. This could be approached through measurement of the fatty acids present in solution over the time of the assay and correlating this to the antimicrobial effect, as this would indicate how the concentration in solution changes over time. Understanding the mechanism of action may also elucidate how the capsule has a protective effect against fish oil treatments. Future work should also investigate variation in efficacy in relation to fish oil source, and pathogen virulence, as well as whether bacteria are able to gain resistance mechanisms against the treatments.

## 5. Conclusions

The results of this study have shown that both fish oil components and manuka honey have the potential to be utilised as treatments in combatting and preventing infection, particularly at a time when antibiotics are becoming less effective. EPA is more effective than DHA, and although the resolvins inhibited *E. coli*, there was no broad-spectrum effect on the other organisms. Manuka honey has shown variation in its efficacy depending on its origin of supply, but both varieties showed positive effects in inhibiting bacterial growth; however, only one had bactericidal and antibiofilm properties.

The results demonstrate that manuka honey has a broad-spectrum effect on bacteria, corroborating results from previous studies. EPA and DHA are also able to treat both Gram-positive and negative bacteria, with the exceptions being the capsuled organisms and the carbapenem-resistant strain of *E. coli.* High-content MGO honey, EPA, and DHA also have the potential to treat biofilms, including ESKAPE and drug-resistant pathogens, and certainly deserve further investigations into their potential uses as treatments for infection.

## Figures and Tables

**Figure 1 microorganisms-12-00778-f001:**
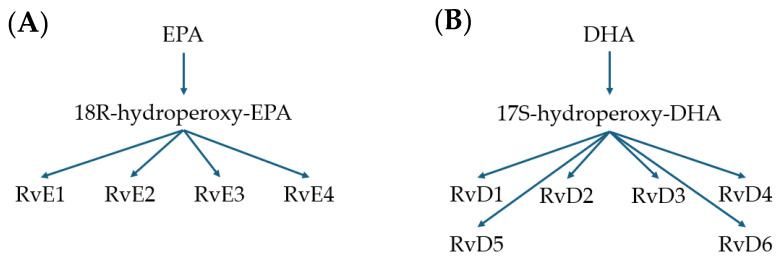
Pathways demonstrating the starting materials of (**A**) eicosapentaenoic acid (EPA) and (**B**) docosahexaenoic acid (DHA) and their intermediates in synthesising resolvins.

**Figure 2 microorganisms-12-00778-f002:**
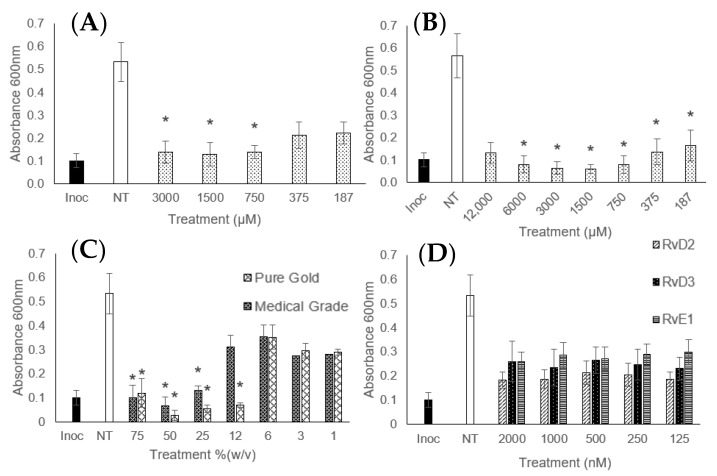
Measured absorbance of MRSA treated with (**A**) docosahexaenoic acid (DHA), (**B**) eicosapentaenoic acid (EPA), (**C**) manuka honey, and (**D**) resolvins. The black ‘inoc’ bar represents the initially measured absorbance at 0 h, and the white ‘NT’ is no treatment—the absorbance of the untreated control after 18 h. Treatments of interest that proved to not be significantly different from the inoculation value are marked with a *. Error bars of ± 1 standard deviation are included, *n* = 9.

**Figure 3 microorganisms-12-00778-f003:**
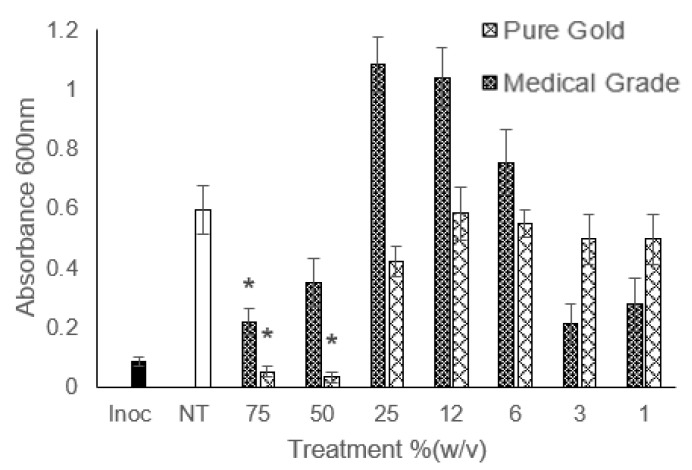
Measured absorbance for MIC assay of *P. aeruginosa* treated with pure gold and medical-grade honey. ‘Inoc’ = Inoculation, ‘NT’ = No Treatment. Treatments of interest that proved to not be significantly different from the inoculation value are marked with a *.

**Table 1 microorganisms-12-00778-t001:** The recorded minimum inhibitory concentrations (MIC), minimum bactericidal concentrations (MBC), and minimum biofilm eradication concentration (MBEC) for all organisms and individual fish oil component treatments. Assays were performed on three separate occasions in triplicate (*n* = 9). Partial inhibition is notated as PI and is highlighted in amber, NBA is ‘No Bactericidal Activity’, and NIA is ‘No Inhibitory Activity’, and these are both highlighted in red. Successful inhibition or bactericidal effects are highlighted in green. The instances where a test was not carried out are indicated by a dash in a grey cell.

	Treatment											
	**DHA (µM)**			**EPA (µM)**			**RvD2 (nM)**		**RvD3 (nM)**		**RvE1 (nM)**	
	**MIC**	**MBC**	**MBEC**	**MIC**	**MBC**	**MBEC**	**MIC**	**MBC**	**MIC**	**MBC**	**MIC**	**MBC**
** *S. epidermidis* **	1500	6000	–	375	3000	–	PI 31	–	PI 125	–	PI 125	–
** *S. aureus* **	375	3000	–	375	1500	–	PI 31	–	PI 125	–	PI 125	–
**MRSA**	750	6000	24,000	187	1500	6000	31	NBA	PI 125	–	PI 125	–
** *E. coli* **	0.25	12,000	24,000	375	3000	6000	125	NBA	125	NBA	125	NBA
**CRE**	PI 187	–	–	PI 187	–	–	PI 125	–	PI 125	–	PI 125	–
** *K. pneumoniae* **	NIA	–	NBA	PI 375	–	NBA	NIA	–	PI 2000	–	PI 2000	–
** *P. aeruginosa* **	PI 187	–	–	PI 187	–	–	NIA	–	PI 125	–	NIA	–
	**Pure Gold Honey (%*w*/*v*)**			**Medical-Grade Honey (%*w*/*v*)**			**Pure Gold MGO Equivalent (%*w*/*v*)**		**Medical-Grade MGO Equivalent (%*w*/*v*)**			
**Organism**	**MIC**	**MBC**	**MBEC**	**MIC**	**MBC**		**MIC**	**MBC**	**MIC**	**MBC**		
** *S. epidermidis* **	12.5	25	–	25	75		12.5	–	25	–		
**MRSA**	12.5	25	75	25	NBA		12.5	12.5	25	50		
** *K. pneumoniae* **	50	50	50	50	NBA		50	–	50	–		
** *S. aureus* **	12.5	25	–	25	NBA		12.5	–	25	–		
** *E. coli* **	50	75	50	50	NBA		50	–	50	–		
**CRE**	50	50	–	50	NBA		50	50	50	75		
** *P. aeruginosa* **	50	75	–	75	NBA		50	–	75	–		

## Data Availability

Data are available on request from the corresponding author.

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
