# Peer review of "The Antimicrobial and Antibiofilm Abilities of Fish Oil Derived Polyunsaturated Fatty Acids and Manuka Honey"

_microorganisms, 2024, doi:10.3390/microorganisms12040778_

Round 1

Reviewer 1 Report

Comments and Suggestions for Authors

Dear Authors,

In this article you present an interesting study about the antibacterial properties of some compounds from fish oil and honey, compounds that can be alternatives in numerous infections with bacterial agents with high pathogenicity, especially in the conditions of the escalation of resistance to antibiotics and last but not least, the ability to greater adaptation of some bacterial species to various antibiotics.

For more clarity of the text and for a clearer picture, some changes or additions are necessary to contribute to an increase in the understanding of this study, which I mention below:

Lines 16-19 - Mentioning for the first time a species of microorganism requires the full name, subsequently abbreviations are used.

Line 36 - I suggest the authors to add to the description in this sentence, the fact that these fatty acids introduced through food that are not synthesized in the body, are for this reason considered essential fatty acids, considering their multiple role in the functioning of the body.

Lines 41-45 - for a clearer picture, apart from the description in the text, but also to emphasize the importance of these essential transformations that take place in the body, I suggest that the authors schematically present the synthesis stages presented.

Line 49 - the abbreviation ESKAPE needs explanation, being the first time mentioned in this context.

Lines 58-60 - I suggest adding more recent studies to support the importance of Manuka honey's methylglyoxal content, which makes it different from other varieties.

Line 75 - I suggest the addition of more recent bibliographic titles than the one presented [19 from 2003], which would emphasize the importance of biofilm formation, the extremely important stage in its eradication.

Line 134 - in order to avoid confusion in the understanding of the applied technique, the statement "MBC was determined visually" requires a short additional explanation.

Line 157 - the mention "Additionally, K. pneumoniae was included due to the variation in results shown 157 among existing literature regarding DHA treatment of these biofilms", requires a citation for support.

The Results section is presented in a clear, explicit and concise manner.

The Discussion section is also remarkable, the authors present both the limitations and the positive results of their study, in a realistic manner. The suggestion that EPA and DHA may be effective therapeutic alternatives in the prevention of bacterial biofilm, indeed requires further study.

Line 306 - The statement that bacteria do not develop resistance mechanisms against these studied compounds, requires additional citation for support.

Finally, in the "authors' contributions" section, I suggest the authors to personalize each author's contribution to this study.

Reviewer 2 Report

Comments and Suggestions for Authors

After reviewing the manuscript titled "The Antimicrobial and Antibiofilm Abilities of Fish Oil-Derived Polyunsaturated Fatty Acids and Manuka Honey" by Jenna Clare, Martin R. Lindley, and Elizabeth Ratcliffe, I have some comments.

I find the evaluation of the MIC and MBC from the fish oils and honey fatty acids to be intriguing. However, this response is not supported by any characteristics of the substances. Theoretically, in the literature, we can find some compositions of the different compounds. However, it is necessary to establish the presence of the fatty acids present in the sample and correlate them with the antibacterial responses. It is well-known that depending on the origin of the fish or honey, the composition could vary. Correlating the concentration of fatty acids with the antibacterial activity could provide insights into the mechanisms of action of these compounds on microorganisms.

Comments on the Quality of English Language

Moderate editing of English language required

Round 2

Reviewer 1 Report

Comments and Suggestions for Authors

Dear Authors,

Thank you for your response to my suggestions. The article revised in this way is much more visible and useful for further research.

Kind regards

Reviewer 2 Report

Comments and Suggestions for Authors

Dear authors.

Thank you for the response and the clarifications. I consider that the modifications to the manuscript are adequate and It is suitable for publication.